# Meta-Inverse Reinforcement Learning with Probabilistic Context Variables

**Lantao Yu**∗, **Tianhe Yu**∗, **Chelsea Finn, Stefano Ermon**
Department of Computer Science, Stanford University
Stanford, CA 94305
{lantaoyu,tianheyu,cbfinn,ermon}@cs.stanford.edu

## Abstract

Providing a suitable reward function to reinforcement learning can be difficult in many real world applications. While inverse reinforcement learning (IRL) holds promise for automatically learning reward functions from demonstrations, several major challenges remain. First, existing IRL methods learn reward functions from scratch, requiring large numbers of demonstrations to correctly infer the reward for each task the agent may need to perform. Second, existing methods typically assume homogeneous demonstrations for a single behavior or task, while in practice, it might be easier to collect datasets of heterogeneous but related behaviors. To this end, we propose a deep latent variable model that is capable of learning rewards from demonstrations of distinct but related tasks in an unsupervised way. Critically, our model can infer rewards for new, structurally-similar tasks from a single demonstration. Our experiments on multiple continuous control tasks demonstrate the effectiveness of our approach compared to state-of-the-art imitation and inverse reinforcement learning methods.

## 1   Introduction

While reinforcement learning (RL) has been successfully applied to a range of decision-making and control tasks in the real world, it relies on a key assumption: having access to a well-defined reward function that measures progress towards the completion of the task. Although it can be straightforward to provide a high-level description of success conditions for a task, existing RL algorithms usually require a more informative signal to expedite exploration and learn complex behaviors in a reasonable time. While reward functions can be hand-specified, reward engineering can require significant human effort. Moreover, for many real-world tasks, it can be challenging to manually design reward functions that actually benefit RL training, and reward mis-specification can hamper autonomous learning [2].

Learning from demonstrations [31] sidesteps the reward specification problem by instead learning directly from expert demonstrations, which can be obtained through teleoperation [39] or from humans experts [38]. Demonstrations can often be easier to provide than rewards, as humans can complete many real-world tasks quite efficiently. Two major methodologies of learning from demonstrations include imitation learning and inverse reinforcement learning. Imitation learning is simple and often exhibits good performance [39, 16]. However, it lacks the ability to transfer learned policies to new settings where the task specification remains the same but the underlying environment dynamics change. As the reward function is often considered as the most succinct, robust and transferable representation of a task [1, 11], the problem of inferring reward functions from expert demonstrations, *i.e.* inverse RL (IRL) [23], is important to consider.

---

∗Equal contribution.

While appealing, IRL still typically relies on large amounts of high-quality expert data, and it can be prohibitively expensive to collect demonstrations that cover all kinds of variations in the wild (*e.g.* opening all kinds of doors or navigating to all possible target positions). As a result, these methods are data-inefficient, particularly when learning rewards for individual tasks in isolation, starting from scratch. On the other hand, meta-learning [32, 4], also known as learning to learn, seeks to exploit the structural similarity among a distribution of tasks and optimizes for rapid adaptation to unknown settings with a limited amount of data. As the reward function is able to succinctly capture the structure of a reinforcement learning task, *e.g.* the goal to achieve, it is promising to develop methods that can quickly infer the structure of a new task, *i.e.* its reward, and train a policy to adapt to it. Xu et al. [36] and Gleave and Habryka [12] have proposed approaches that combine IRL and gradient-based meta-learning [9], which provide promising results on deriving generalizable reward functions. However, they have been limited to tabular MDPs [36] or settings with provided task distributions [12], which are challenging to gather in real-world applications.

The primary contribution of this paper is a new framework, termed Probabilistic Embeddings for Meta-Inverse Reinforcement Learning (PEMIRL), which enables meta-learning of rewards from *unstructured* multi-task demonstrations. In particular, PEMIRL combines and integrates ideas from context-based meta-learning [5, 26], deep latent variable generative models [17], and maximum entropy inverse RL [42, 41], into a unified graphical model (see Figure 4 in Appendix D) that bridges the gap between few-shot reward inference and learning from unstructured, heterogeneous demonstrations. PEMIRL can learn robust reward functions that generalize to new tasks with a *single* demonstration on complex domains with continuous state-action spaces, while meta-training on a set of unstructured demonstrations without specified task groupings or labeling for each demonstration. Our experiment results on various continuous control tasks including Point-Maze, Ant, Sweeper, and Sawyer Pusher demonstrate the effectiveness and scalability of our method.

## 2 Preliminaries

**Markov Decision Process (MDP).** A discrete-time finite-horizon MDP is defined by a tuple $(T, \mathcal{S}, \mathcal{A}, P, r, \eta)$, where $T$ is the time horizon; $\mathcal{S}$ is the state space; $\mathcal{A}$ is the action space; $P : \mathcal{S} \times \mathcal{A} \times \mathcal{S} \to [0, 1]$ describes the (stochastic) transition process between states; $r : \mathcal{S} \times \mathcal{A} \to \mathbb{R}$ is a bounded reward function; $\eta \in \mathcal{P}(\mathcal{S})$ specifies the initial state distribution, where $\mathcal{P}(\mathcal{S})$ denotes the set of probability distributions over the state space $\mathcal{S}$. We use $\tau$ to denote a trajectory, *i.e.* a sequence of state action pairs for one episode. We also use $\rho_\pi(s_t)$ and $\rho_\pi(s_t, a_t)$ to denote the state and state-action marginal distribution encountered when executing a policy $\pi(a_t|s_t)$.

**Maximum Entropy Inverse Reinforcement Learning (MaxEnt IRL).** The maximum entropy reinforcement learning (MaxEnt RL) objective is defined as:

$$\max_\pi \sum_{t=1}^T \mathbb{E}_{(s_t,a_t) \sim \rho_\pi} [r(s_t, a_t) + \alpha \mathcal{H}(\pi(\cdot|s_t))] \tag{1}$$

which augments the reward function with a causal entropy regularization term $\mathcal{H}(\pi) = \mathbb{E}_\pi[-\log \pi(a|s)]$. Here $\alpha$ is an optional parameter to control the relative importance of reward and entropy. For notational simplicity, without loss of generality, in the following we will assume $\alpha = 1$. Given some expert policy $\pi_E$ that is obtained by above MaxEnt RL procedure , the MaxEnt IRL framework [42] aims to find a reward function that rationalizes the expert behaviors, which can be interpreted as solving the following maximum likelihood estimation (MLE) problem:

$$p_\theta(\tau) \propto \left[ \eta(s_1) \prod_{t=1}^T P(s_{t+1}|s_t, a_t) \right] \exp\left( \sum_{t=1}^T r_\theta(s_t, a_t) \right) = \overline{p_\theta}(\tau) \tag{2}$$

$$\arg\min_\theta D_{\mathrm{KL}}(p_{\pi_E(\tau)} || p_\theta(\tau)) = \arg\max_\theta \mathbb{E}_{p_{\pi_E}(\tau)}[\log p_\theta(\tau)] = \mathbb{E}_{\tau \sim \pi_E} \left[ \sum_{t=1}^T r_\theta(s_t, a_t) \right] - \log Z_\theta$$

Here, $\theta$ is the parameter of the reward function and $Z_\theta$ is the partition function, *i.e.* $\int \overline{p_\theta}(\tau) d\tau$, an integral over all possible trajectories consistent with the environment dynamics. $Z_\theta$ is intractable to compute when state-action spaces are large or continuous, or environment dynamics are unknown.

Finn et al. [7] and Fu et al. [11] proposed the adversarial IRL (AIRL) framework as an efficient sampling-based approximation to MaxEnt IRL, which resembles Generative Adversarial Networks [13]. Specially, in AIRL, there is a discriminator $D_\theta$ (a binary classifier) parametrized by $\theta$ and an adaptive sampler $\pi_\omega$ (a policy) parametrized by $\omega$. The discriminator takes a particular form: $D_\theta(s, a) = \exp(f_\theta(s, a))/(\exp(f_\theta(s, a)) + \pi_\omega(a|s))$, where $f_\theta(s, a)$ is the learned reward function and $\pi_\omega(a|s)$ is pre-computed as an input to the discriminator. The discriminator is trained to distinguish between the trajectories sampled from the expert and the adaptive sampler; while the adaptive sampler $\pi_\omega(a|s)$ is trained to maximize $\mathbb{E}_{\rho_{\pi_\omega}}[\log D_\theta(s, a) - \log(1 - D_\theta(s, a))]$, which is equivalent to maximizing the following entropy regularized policy objective (with $f_\theta(s, a)$ serving as the reward function):

$$\mathbb{E}_{\pi_\omega}\left[\sum_{t=1}^{T} \log(D_\theta(s_t, a_t)) - \log(1 - D_\theta(s_t, a_t))\right] = \mathbb{E}_{\pi_\omega}\left[\sum_{t=1}^{T} f_\theta(s_t, a_t) - \log \pi_\omega(a_t|s_t)\right] \quad (3)$$

Under certain conditions, it can be shown that the learned reward function will recover the ground-truth reward up to a constant (Theorem C.1 in [11]).

## 3 Probabilistic Embeddings for Meta-Inverse Reinforcement Learning

### 3.1 Problem Statement

Before defining our meta-inverse reinforcement learning problem (Meta-IRL), we first define the concept of optimal context-conditional policy.

We start by generalizing the notion of MDP with a probabilistic context variable denoted as $m \in \mathcal{M}$, where $\mathcal{M}$ is the (discrete or continuous) value space of $m$. For example, in a navigation task, the context variables could represent different goal positions in the environment. Now, each component of the MDP has an additional dependency on the context variable $m$. For example, by slightly overloading the notation, the reward function is now defined as $r : \mathcal{S} \times \mathcal{A} \times \mathcal{M} \to \mathbb{R}$. For simplicity, the state space, action space, initial state distribution and transition dynamics are often assumed to be independent of $m$ [5, 9], which we will follow in this work. Intuitively, different $m$'s correspond to different tasks with shared structures.

Given above definitions, the context-conditional trajectory distribution induced by a context-conditional policy $\pi : \mathcal{S} \times \mathcal{M} \to \mathcal{P}(\mathcal{A})$ can be written as:

$$p_\pi(\tau = \{\boldsymbol{s}_{1:T}, \boldsymbol{a}_{1:T}\}|m) = \eta(s_1) \prod_{t=1}^{T} \pi(\boldsymbol{a}_t|s_t, m) P(s_{t+1}|s_t, a_t) \quad (4)$$

Let $p(m)$ denote the prior distribution of the latent context variable (which is a part of the problem definition). With the conditional distribution defined above, the optimal entropy-regularized context-conditional policy is defined as:

$$\pi^* = \arg\max_\pi \mathbb{E}_{m \sim p(m), (\boldsymbol{s}_{1:T}, \boldsymbol{a}_{1:T}) \sim p_\pi(\cdot|m)}\left[\sum_{t=1}^{T} r(s_t, a_t, m) - \log \pi(a_t|s_t, m)\right] \quad (5)$$

Now, let us introduce the problem of Meta-IRL from heterogeneous multi-task demonstration data. Suppose there is some ground-truth reward function $r(s, a, m)$ and a corresponding expert policy $\pi_E(a_t|s_t, m)$ obtained by solving the optimization problem defined in Equation (5). Given a set of demonstrations *i.i.d.* sampled from the induced marginal distribution $p_{\pi_E}(\tau) = \int_\mathcal{M} p(m) p_{\pi_E}(\tau|m) dm$, the goal is to meta-learn an inference model $q(m|\tau)$ and a reward function $f(s, a, m)$, such that given some new demonstration $\tau_E$ generated by sampling $m' \sim p(m), \tau_E \sim p_{\pi_E}(\tau|m')$, with $\hat{m}$ being inferred as $\hat{m} \sim q(m|\tau_E)$, the learned reward function $f(s, a, \hat{m})$ and the ground-truth reward $r(s, a, m')$ will induce the same set of optimal policies [24].

Critically, we assume no knowledge of the prior task distribution $p(m)$, the latent context variable $m$ associated with each demonstration, nor the transition dynamics $P(s_{t+1}|s_t, a_t)$ during meta-training. Note that the entire supervision comes from the provided unstructured demonstrations, which means we also do not assume further interactions with the experts as in Ross et al. [28].

## 3.2   Meta-IRL with Mutual Information Regularization over Context Variables

Under the framework of MaxEnt IRL, we first parametrize the context variable inference model $q_\psi(m|\tau)$ and the reward function $f_\theta(s, a, m)$ (where the input $m$ is inferred by $q_\psi$), The induced $\theta$-parametrized trajectory distribution is given by:

$$p_\theta(\tau = \{\boldsymbol{s}_{1:T}, \boldsymbol{a}_{1:T}\}|m) = \frac{1}{Z(\theta)} \left[\eta(s_1) \prod_{t=1}^{T} P(s_{t+1}|s_t, a_t)\right] \exp\left(\sum_{t=1}^{T} f_\theta(s_t, a_t, m)\right) \quad (6)$$

where $Z(\theta)$ is the partition function, *i.e.*, an integral over all possible trajectories. Without further constraints over $m$, directly applying AIRL to learning the reward function (by augmenting each component of AIRL with an additional context variable $m$ inferred by $q_\psi$) could simply ignore $m$, which is similar to the case of InfoGAIL [21]. Therefore, some connection between the reward function and the latent context variable $m$ need to be established. With MaxEnt IRL, a parametrized reward function will induce a trajectory distribution. From the perspective of information theory, the mutual information between the context variable $m$ and the trajectories sampled from the reward induced distribution will provide an ideal measure for such a connection.

Formally, the mutual information between two random variables $m$ and $\tau$ under joint distribution $p_\theta(m, \tau) = p(m)p_\theta(\tau|m)$ is given by:

$$I_{p_\theta}(m; \tau) = \mathbb{E}_{m \sim p(m), \tau \sim p_\theta(\tau|m)}[\log p_\theta(m|\tau) - \log p(m)] \quad (7)$$

where $p_\theta(\tau|m)$ is the conditional distribution (Equation (6)), and $p_\theta(m|\tau)$ is the corresponding posterior distribution.

As we do not have access to the prior distribution $p(m)$ and posterior distribution $p_\theta(m|\tau)$, directly optimizing the mutual information in Equation (7) is intractable. Fortunately, we can leverage $q_\psi(m|\tau)$ as a variational approximation to $p_\theta(m|\tau)$ to reason about the uncertainty over tasks, as well as conduct approximate sampling from $p(m)$ (we will elaborate this later in Section 3.3). Formally, let $p_{\pi_E}(\tau)$ denote the expert trajectory distribution, we have the following desiderata:

**Desideratum 1**. Matching conditional distributions: $\mathbb{E}_{p(m)}\left[D_{\mathrm{KL}}(p_{\pi_E}(\tau|m)||p_\theta(\tau|m))\right] = 0$

**Desideratum 2**. Matching posterior distributions: $\mathbb{E}_{p_\theta(\tau)}[D_{\mathrm{KL}}(p_\theta(m|\tau)||q_\psi(m|\tau))] = 0$

The first desideratum will encourage the $\theta$-induced conditional trajectory distribution to match the empirical distribution implicitly defined by the expert demonstrations, which is equivalent to the MLE objective in the MaxEnt IRL framework. Note that they also share the same marginal distribution over the context variable $p(m)$, which implies that matching the conditionals in Desideratum 1 will also encourage the joint distributions, conditional distributions $p_{\pi_E}(m|\tau)$ and $p_\theta(m|\tau)$, and marginal distributions over $\tau$ to be matched. The second desideratum will encourage the variational posterior $q_\psi(m|\tau)$ to be a good approximation to $p_\theta(m|\tau)$ such that $q_\psi(m|\tau)$ can correctly infer the latent context variable given a new expert demonstration sampled from a new task.

With the mutual information (Equation (7)) being the objective, and Desideratum 1 and 2 being the constraints, the meta-inverse reinforcement learning with probabilistic context variables problem can be interpreted as a constrained optimization problem, whose Lagrangian dual function is given by:

$$\min_{\theta,\psi} -I_{p_\theta}(m; \tau) + \alpha \cdot \mathbb{E}_{p(m)}\left[D_{\mathrm{KL}}(p_{\pi_E}(\tau|m)||p_\theta(\tau|m))\right] + \beta \cdot \mathbb{E}_{p_\theta(\tau)}[D_{\mathrm{KL}}(p_\theta(m|\tau)||q_\psi(m|\tau))] \quad (8)$$

With the Lagrangian multipliers taking specific values ($\alpha = 1, \beta = 1$) [40], the above Lagrangian dual function can be rewritten as:

$$\min_{\theta,\psi} \mathbb{E}_{p(m)}\left[D_{\mathrm{KL}}(p_{\pi_E}(\tau|m)||p_\theta(\tau|m))\right] + \mathbb{E}_{p_\theta(m,\tau)}\left[\log \frac{p(m)}{p_\theta(m|\tau)} + \log \frac{p_\theta(m|\tau)}{q_\psi(m|\tau)}\right]$$

$$\equiv \max_{\theta,\psi} -\mathbb{E}_{p(m)}\left[D_{\mathrm{KL}}(p_{\pi_E}(\tau|m)||p_\theta(\tau|m))\right] + \mathbb{E}_{m \sim p(m), \tau \sim p_\theta(\tau|m)}[\log q_\psi(m|\tau)] \quad (9)$$

$$= \max_{\theta,\psi} -\mathbb{E}_{p(m)}\left[D_{\mathrm{KL}}(p_{\pi_E}(\tau|m)||p_\theta(\tau|m))\right] + \mathcal{L}_{\mathrm{info}}(\theta, \psi) \quad (10)$$

Here the negative entropy term $-H_p(m) = \mathbb{E}_{p_\theta(m,\tau)}[\log p(m)] = \mathbb{E}_{p(m)}[\log p(m)]$ is omitted (in Eq. (9)) as it can be treated as a constant in the optimization procedure of parameters $\theta$ and $\psi$.

## 3.3 Achieving Tractability with Sampling-Based Gradient Estimation

Note that Equation (10) cannot be evaluated directly, as the first term requires estimating the KL divergence between the empirical expert distribution and the energy-based trajectory distribution $p_\theta(\tau|m)$ (induced by the $\theta$-parametrized reward function), and the second term requires sampling from it. For the purpose of optimizing the first term in Equation (10), as introduced in Section 2, we can employ the adversarial reward learning framework [11] to construct an efficient sampling-based approximation to the maximum likelihood objective. Note that different from the original AIRL framework, now the adaptive sampler $\pi_\omega(a|s, m)$ is additionally conditioned on the context variable $m$. Furthermore, we here introduce the following lemma, which will be helpful for deriving the optimization of the second term in Equation (10).

**Lemma 1.** *In context variable augmented Adversarial IRL (with the adaptive sampler being $\pi_\omega(a|s, m)$ and the discriminator being $D_\theta(s, a, m) = \frac{\exp(f_\theta(s, a, m))}{\exp(f_\theta(s, a, m)) + \pi_\omega(a|s, m)})$ , under deterministic dynamics, when training the adaptive sampler $\pi_\omega$ with reward signal $(\log D_\theta - \log(1 - D_\theta))$ to optimality, the trajectory distribution induced by $\pi_\omega^*$ corresponds to the maximum entropy trajectory distribution with $f_\theta(s, a, m)$ serving as the reward function:*

$$p_{\pi_\omega^*}(\tau|m) = \frac{1}{Z_\theta}\left[\eta(s_1)\prod_{t=1}^{T} P(s_{t+1}|s_t, a_t)\right]\exp\left(\sum_{t=1}^{T} f_\theta(s_t, a_t, m)\right) = p_\theta(\tau|m)$$

*Proof.* See Appendix A. □

Now we are ready to introduce how to approximately optimize the second term of the objective in Equation (10) w.r.t. $\theta$ and $\psi$. First, we observe that the gradient of $\mathcal{L}_{\text{info}}(\theta, \psi)$ w.r.t. $\psi$ is given by:

$$\frac{\partial}{\partial\psi}\mathcal{L}_{\text{info}}(\theta, \psi) = \mathbb{E}_{m\sim p(m), \tau\sim p_\theta(\tau|m)}\frac{1}{q(m|\tau, \psi)}\frac{\partial q(m|\tau, \psi)}{\partial\psi} \tag{11}$$

Thus to construct an estimate of the gradient in Equation (11), we need to obtain samples from the $\theta$-induced trajectory distribution $p_\theta(\tau|m)$. With Lemma 1, we know that when the adaptive sampler $\pi_\omega$ in AIRL is trained to optimality, we can use $\pi_\omega^*$ to construct samples, as the trajectory distribution $p_{\pi_\omega^*}(\tau|m)$ matches the desired distribution $p_\theta(\tau|m)$.

Also note that the expectation in Equation (11) is also taken over the prior task distribution $p(m)$. In cases where we have access to the ground-truth prior distribution, we can directly sample $m$ from it and use $p_{\pi_\omega^*}(\tau|m)$ to construct a gradient estimation. For the most general case, where we do not have access to $p(m)$ but instead have expert demonstrations sampled from $p_{\pi_E}(\tau)$, we use the following generative process:

$$\tau \sim p_{\pi_E(\tau)}, m \sim q_\psi(m|\tau) \tag{12}$$

to synthesize latent context variables, which approximates the prior task distribution when $\theta$ and $\psi$ are trained to optimality.

To optimize $\mathcal{L}_{\text{info}}(\theta, \psi)$ w.r.t. $\theta$, which is an important step of updating the reward function parameters such that it encodes the information of the latent context variable, different from the optimization of Equation (11), we cannot directly replace $p_\theta(\tau|m)$ with $p_{\pi_\omega}(\tau|m)$. The reason is that we can only use the approximation of $p_\theta$ to do inference (*i.e.* computing the value of an expectation). When we want to optimize an expectation ($\mathcal{L}_{\text{info}}(\theta, \psi)$) w.r.t. $\theta$ and the expectation is taken over $p_\theta$ itself, we cannot instead replace $p_\theta$ with $\pi_\omega$ to do the sampling for estimating the expectation. In the following, we discuss how to estimate the gradient of $\mathcal{L}_{\text{info}}(\theta, \psi)$ w.r.t. $\theta$ with empirical samples from $\pi_\omega$.

**Lemma 2.** *The gradient of $\mathcal{L}_{info}(\theta, \psi)$ w.r.t. $\theta$ can be estimated with:*

$$\mathbb{E}_{m\sim p(m), \tau\sim p_{\pi_\omega^*}(\tau|m)}\left[\log q_\psi(m|\tau)\left[\sum_{t=1}^{T}\frac{\partial}{\partial\theta}f_\theta(s_t, a_t, m) - \mathbb{E}_{\tau'\sim p_{\pi_\omega^*}(\tau|m)}\sum_{t=1}^{T}\frac{\partial}{\partial\theta}f_\theta(s'_t, a'_t, m)\right]\right]$$

*When $\omega$ is trained to optimality, the estimation is unbiased.*

*Proof.* See Appendix B. □

With Lemma 2, as before, we can use the generative process in Equation (12) to sample $m$ and use the conditional trajectory distribution $p_{\pi_\omega^*}(\tau|m)$ to sample trajectories for estimating $\frac{\partial}{\partial\theta}\mathcal{L}_{\text{info}}(\theta, \psi)$. The overall training objective of PEMIRL is:

$$\min_\omega \max_{\theta,\psi} \mathbb{E}_{\tau_E \sim p_{\pi_E}(\tau), m \sim q_\psi(m|\tau_E), (s,a) \sim \rho_{\pi_\omega}(s,a|m)} \log(1 - D_\theta(s, a, m)) +$$

$$\mathbb{E}_{\tau_E \sim p_{\pi_E}(\tau), m \sim q_\psi(m|\tau_E)} \log(D_\theta(s, a, m)) + \mathcal{L}_{\text{info}}(\theta, \psi) \qquad (13)$$

$$\text{where } D_\theta(s, a, m) = \exp(f_\theta(s, a, m))/(\exp(f_\theta(s, a, m)) + \pi_\omega(a|s, m))$$

We summarize the meta-training procedure in Algorithm 1 and the meta-test procedure in Appendix C.

---

**Algorithm 1** PEMIRL Meta-Training

---

**Input:** Expert trajectories $\mathcal{D}_E = \{\tau_E^j\}$; Initial parameters of $f_\theta, \pi_\omega, q_\psi$.
**repeat**
    Sample two batches of unlabeled demonstrations: $\tau_E, \tau_E' \sim \mathcal{D}_E$
    Infer a batch of latent context variables from the sampled demonstrations: $m \sim q_\psi(m|\tau_E)$
    Sample trajectories $\mathcal{D}$ from $\pi_\omega(\tau|m)$, with the latent context variable fixed during each rollout and included in $\mathcal{D}$.
    Update $\psi$ to increase $\mathcal{L}_{\text{info}}(\theta, \psi)$ with gradients in Equation (11), with samples from $\mathcal{D}$.
    Update $\theta$ to increase $\mathcal{L}_{\text{info}}(\theta, \psi)$ with gradients in Equation (15), with samples from $\mathcal{D}$.
    Update $\theta$ to decrease the binary classification loss:
        $\mathbb{E}_{(s,a,m) \sim \mathcal{D}}[\nabla_\theta \log D_\theta(s, a, m)] + \mathbb{E}_{\tau_E' \sim \mathcal{D}_E, m \sim q_\psi(m|\tau_E')}[\nabla_\theta \log(1 - D_\theta(s, a, m))]$
    Update $\omega$ with TRPO to increase the following objective: $\mathbb{E}_{(s,a,m) \sim \mathcal{D}}[\log D_\theta(s, a, m)]$
**until** Convergence
**Output:** Learned inference model $q_\psi(m|\tau)$, reward function $f_\theta(s, a, m)$ and policy $\pi_\omega(a|s, m)$.

---

## 4 Related Work

Inverse reinforcement learning (IRL), first introduced by Ng and Russell [23], is the problem of learning reward functions directly from expert demonstrations. Prior work tackling IRL include margin-based methods [1, 27] and maximum entropy (MaxEnt) methods [42]. Margin-based methods suffer from being an underdefined problem, while MaxEnt requires the algorithm to solve the forward RL problem in the inner loop, making it challenging to use in non-tabular settings. Recent works have scaled MaxEnt IRL to large function approximators, such as neural networks, by only partially solving the forward problem in the inner loop, developing an adversarial framework for IRL [7, 8, 11, 25]. Other imitation learning approaches [16, 21, 14, 18] are also based on the adversarial framework, but they do not recover a reward function. We build upon the ideas in these single-task IRL works. Instead of considering the problem of learning reward functions for a single task, we aim at the problem of inferring a reward that is disentangled from the environment dynamics and can quickly adapt to new tasks from a single demonstration by leveraging prior data.

We base our work on the problem of meta-learning. Prior work has proposed memory-based methods [5, 30, 22, 26] and methods that learn an optimizer and/or a parameter initialization [3, 20, 9]. We adopt a memory-based meta-learning method similar to [26], which uses a deep latent variable generative model [17] to infer different tasks from demonstrations. While prior multi-task and meta-RL methods [15, 26, 29] have investigated the effectiveness of applying latent variable generative models to learning task embeddings, we focus on the IRL problem instead. Meta-IRL [36, 12] incorporates meta-learning and IRL, showing fast adaptation of the reward functions to unseen tasks. Unlike these approaches, our method is not restrictred to discrete tabular settings and does not require access to grouped demonstrations sampled from a task distribution. Meanwhile, one-shot imitation learning [6, 10, 38, 37] demonstrates impressive results on learning new tasks using a single demonstration; yet, they also require paired demonstrations from each task and hence need prior knowledge on the task distribution. More importantly, one-shot imitation learning approaches only recover a policy, and cannot use additional trials to continue to improve, which is possible when a reward function is inferred instead. Several prior approaches for multi-task imitation learning [21, 14, 34] propose to use unstructured demonstrations without knowing the task distribution, but they neither study quick generalization to new tasks nor provide a reward function. Our work is thus driven by the goal of extending meta-IRL to addressing challenging high-dimensional control tasks with the help of an unstructured demonstration dataset.

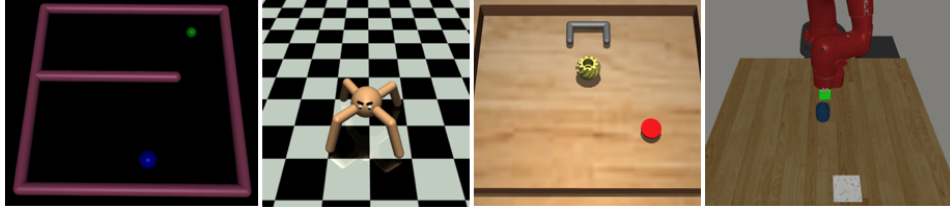

Figure 1: **Experimental domains** (left to right): Point-Maze, Ant, Sweeper, and Sawyer Pusher.

## 5 Experiments

In this section, we seek to investigate the following two questions: (1) Can PEMIRL learn a policy with competitive few-shot generalization abilities compared to one-shot imitation learning methods using only unstructured demonstrations? (2) Can PEMIRL efficiently infer robust reward functions of new continuous control tasks where one-shot imitation learning fails to generalize, enabling an agent to continue to improve with more trials?

We evaluate our method on four simulated domains using the Mujoco physics engine [35]. To our knowledge, there's no prior work on designing meta-IRL or one-shot imitation learning methods for complex domains with high-dimensional continuous state-action spaces with unstructured demonstrations. Hence, we also designed the following variants of existing state-of-the-art (one-shot) imitation learning and IRL methods so that they can be used as fair comparisons to our method:

- **AIRL**: The original AIRL algorithm without incorporating latent context variables, trained across all demonstrations.

- **Meta-Imitation Learning with Latent Context Variables (Meta-IL)**: As in [26], we use the inference model $q_\psi(m|\tau)$ to infer the context of a new task from a single demonstrated trajectory, denoted as $\hat{m}$, and then train the conditional imitaiton policy $\pi_\omega(a|s, \hat{m})$ using the same demonstration. This approach also resembles [6].

- **Meta-InfoGAIL**: Similar to the method above, except that an additional discriminator $D(s, a)$ is introduced to distinguish between expert and sample trajectories, and trained along with the conditional policy using InfoGAIL [21] objective.

We use trust region policy optimization (TRPO) [33] as our policy optimization algorithm across all methods. We collect demonstrations by training experts with TRPO using ground truth reward. However, the ground truth reward is not available to imitation learning and IRL algorithms. We provide full hyperparameters, architecture information, data efficiency, and experimental setup details in Appendix F. We also include ablation studies on sensitivity of the latent dimensions, importance of the mutual information objective and the performance on stochastic environments in Appendix E. Full video results are on the anonymous supplementary website[2] and our code is open-sourced on GitHub[3].

### 5.1 Policy Performance on Test Tasks

We first answer our first question by showing that our method is able to learn a policy that can adapt to test tasks from a single demonstration, on four continuous control domains: **Point Maze Navigation**: In this domain, a pointmass needs to navigate around a barrier to reach the goal. Different tasks correspond to different goal positions and the reward function measures the distance between the pointmass and the goal position; **Ant**: Similar to [9], this locomotion task requires fast adaptation to walking directions of the ant where the ant needs to learn to move backward or forward depending on the demonstration; **Sweeper**: A robot arm needs to sweep an object to a particular goal position. Fast adaptation of this domain corresponds to different goal locations in the plane; **Sawyer Pusher**: A simulated Sawyer robot is required to push a mug to a variety of goal positions and generalize to unseen goals. We illustrate the set-up for these experimental domains in Figure 1.

|                | Point Maze        | Ant               | Sweeper            | Sawyer Pusher      |
| -------------- | ----------------- | ----------------- | ------------------ | ------------------ |
| Expert         | $-5.21 \pm 0.93$  | $968.80 \pm 27.11$ | $-50.86 \pm 4.75$  | $-23.36 \pm 2.54$  |
| Random         | $-51.39 \pm 10.31$ | $-55.65 \pm 18.39$ | $-259.71 \pm 11.24$ | $-106.88 \pm 18.06$ |
| AIRL [11]      | $-18.15 \pm 3.17$ | $127.61 \pm 27.34$ | $-152.78 \pm 7.39$ | $-51.56 \pm 8.57$  |
| Meta-IL        | $\mathbf{-6.68} \pm 1.51$ | $218.53 \pm 26.48$ | $-89.02 \pm 7.06$  | $-28.13 \pm 4.93$  |
| Meta-InfoGAIL  | $-7.66 \pm 1.85$  | $\mathbf{871.93} \pm 31.28$ | $-87.06 \pm 6.57$  | $-27.56 \pm 4.86$  |
| PEMIRL (ours)  | $-7.37 \pm 1.02$  | $846.18 \pm 25.81$ | $\mathbf{-74.17} \pm 5.31$ | $\mathbf{-27.16} \pm 3.11$ |

Table 1: One-shot policy generalization to test tasks on four experimental domains. Average return and standard deviations are reported over 5 runs.

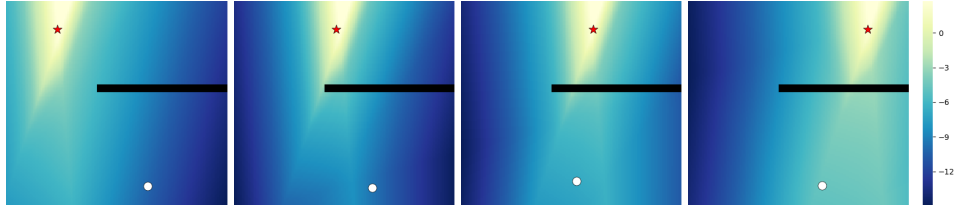

Figure 2: Visualizations of learned reward functions for point-maze navigation. The red star represents the target position and the white circle represents the initial position of the agent (both are different across different iterations). The black horizontal line represents the barrier that cannot be crossed. To show the generalization ability, the expert demonstration used to infer the target position are sampled from new target positions that have not been seen in the meta-training set.

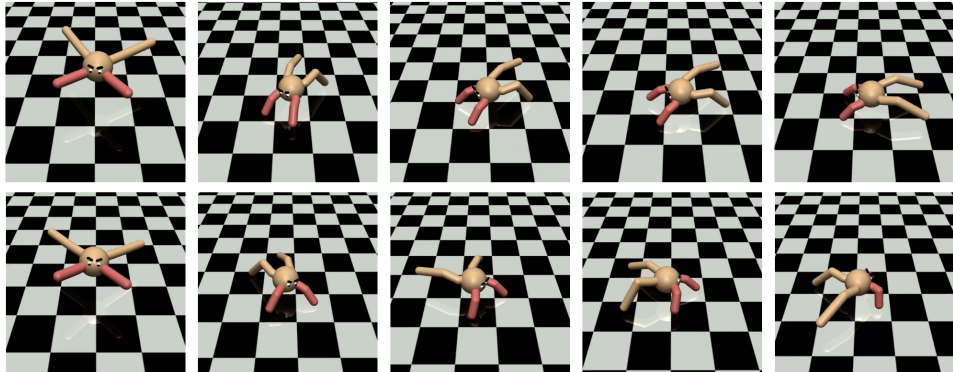

Figure 3: From top to bottom, we show the disabled ant running forward and backward respectively.

We summarize the results in Table 1. PEMIRL achieves comparable imitation performance compared to Meta-IL and Meta-InfoGAIL, while AIRL is incapable of handling multi-task scenarios without incorporating the latent context variables.

## 5.2  Reward Adaptation to Challenging Situations

After demonstrating that the policy learned by our method is able to achieve competitive "one-shot" generalization ability, we now answer the second question by showing PEMIRL learns a robust reward that can adapt to new and more challenging settings where the imitation learning methods and the original AIRL fail. Specifically, after providing the demonstration of an unseen task to the agent, we change the underlying environment dynamics but keep the same task goal. In order to succeed in the task with new dynamics, the agent must correctly infer the underlying goal of the task instead of simply mimicking the demonstration. We show the effectiveness of our reward generalization by training a new policy with TRPO using the learned reward functions on the new task.

| | Method | Point-Maze-Shift | Disabled-Ant |
|---|---|---|---|
| Policy Generalization | Meta-IL | $-28.61 \pm 3.71$ | $-27.86 \pm 10.31$ |
| | Meta-InfoGAIL | $-29.40 \pm 3.05$ | $-51.08 \pm 4.81$ |
| | PEMIRL | $-28.93 \pm 3.59$ | $-46.77 \pm 5.54$ |
| Reward Adaptation | AIRL | $-29.07 \pm 4.12$ | $-76.21 \pm 10.35$ |
| | Meta-InfoGAIL | $-29.72 \pm 3.11$ | $-38.73 \pm 6.41$ |
| | PEMIRL (ours) | $\mathbf{-9.04} \pm 1.09$ | $\mathbf{152.62} \pm 11.75$ |
| | Expert | $-5.37 \pm 0.86$ | $331.17 \pm 17.82$ |

Table 2: Results on direct policy generalization and reward adaptation to challenging situations. Policy generalization examines if the policy learned by Meta-IL is able to generalize to new tasks with new dynamics, while reward adaptation tests if the learned RL can lead to efficient RL training in the same setting. The RL agent learned by PEMIRL rewards outperforms other methods in such challenging settings.

**Point-Maze Navigation with a Shifted Barrier**. Following the setup of Fu et al. [11], at meta-test time, after showing a demonstration moving towards a new target position, we change the position of the barrier from left to right. As the agent must adapt by reaching the target with a different path from what was demonstrated during meta-training, it cannot succeed without correctly inferring the true goal (the target position in the maze) and learning from trial-and-error. As a result, all direct policy generalization approaches fail as all the policies are still directing the pointmass to the right side of the maze. As shown in Figure 2, PEMIRL learns disentangled reward functions that successfully infer the underlying goal of the new task without much reward shaping. Such reward functions enable the RL agent to bypass the right barrier and reach the true goal position. The RL agent trained with the reward learned by AIRL also fail to bypass the barrier and navigate to the target position, as without incorporating the latent context variables and treating the demonstration as multi-modal, AIRL learns an "average" reward and policy among different tasks. We also use the output of the discriminator of Meta-InfoGAIL as reward signals and evaluate its adaptation performance. The agent trained by this reward fails to complete the task since Meta-InfoGAIL does not explicitly optimize for reward learning and the discriminator output converges to uninformative uniform distribution at convergence.

**Disabled Ant Walking**. As in Fu et al. [11], we disable and shorten two front legs of the ant such that it cannot walk without changing its gait to a large extent. Similar to Point-Maze-Shift, all imitaiton policies fail to maneuver the disabled ant to the right direction. As shown in Figure 3, reward functions learned by PEMIRL encourage the RL policy to orient the ant towards the demonstrated direction and move along that direction using two healthy legs, which is only possible when the inferred reward corresponds to the true underlying goal and is disentangled with the dynamics. In contrast, the learned reward of original AIRL as well as the discriminator output of Meta-InfoGAIL cannot infer the underlying goal of the task and provide precise supervision signal, which leads to the unsatisfactory performance of the induced RL policies. Quantitative results are presented in Table 2.

## 6  Conclusion

In this paper, we propose a new meta-inverse reinforcement learning algorithm, PEMIRL, which is able to efficiently infer robust reward functions that are disentangled from the dynamics and highly correlated with the ground-truth rewards under meta-learning settings. To our knowledge, PEMIRL is the first model-free Meta-IRL algorithm that can achieve this and scale to complex domains with continuous state-action spaces. PEMIRL generalizes to new tasks by performing inference over a latent context variable with a single demonstration, on which the recovered policy and reward function are conditioned. Extensive experimental results demonstrate the scalability and effectiveness of our method against strong baselines.

## Acknowledgments

This research was supported by Toyota Research Institute, NSF (#1651565, #1522054, #1733686), ONR (N00014-19-1-2145), AFOSR (FA9550- 19-1-0024). The authors would like to thank Chris Cundy for discussions over the paper draft.

## Footnotes

[2]Video results can be found at: `https://sites.google.com/view/pemirl`

[3]Our implementation of PEMIRL can be found at: `https://github.com/ermongroup/MetaIRL`

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
