[Supplementary Material]

## A Proof of Lemma 1

From Section 2.3 and 2.4 in [19], we know that the policy whose induced trajectory distribution is Equation (6) takes the following energy-based form:

$$\pi_\theta(a_t|s_t, m) = \exp(Q_{\text{soft}}(s_t, a_t, m) - V_{\text{soft}}(s_t, m))$$
$$Q_{\text{soft}}(s_t, a_t, m) = f_\theta(s_t, a_t, m) + \log \mathbb{E}_{s_{t+1}\sim P(\cdot|s_t, a_t, m)}[\exp(V_{\text{soft}}(s_{t+1}, m))]$$
$$V_{\text{soft}}(s_t, m) = \log \int_{\mathcal{A}} \exp(Q_{\text{soft}}(s_t, a', m))da'$$

which corresponds to the optimal policy to the following entropy regularized reinforcement learning problem (for a certain value of $m$):

$$\max_\pi \mathbb{E}_\pi \left[ \sum_{t=1}^{T} f_\theta(s_t, a_t, m) - \log \pi(a_t|s_t, m) \right] \tag{14}$$

From Section 2, we know that Equation (14) is exactly the training objective for the adaptive sampler $\pi_\omega$ in AIRL. Thus, the trajectory distribution of the optimal policy $\pi_{\omega^*}$ matches $p_\theta(\tau|m)$ defined in Equation (6).

## B Proof of Lemma 2

First, the gradient of $\mathcal{L}_{\text{info}}(\theta, \psi)$ w.r.t. $\theta$ can be written as:

$$\frac{\partial}{\partial \theta} \mathcal{L}_{\text{info}}(\theta, \psi) = \mathbb{E}_{m\sim p(m), \tau\sim p(\tau|m, \theta)} \log q(m|\tau, \psi) \frac{\partial}{\partial \theta} \log p_\theta(\tau|m) \tag{15}$$

As $p_\theta(\tau|m)$ is an energy-based distribution (Equation (6)), we need to derive the gradient of $\log p(\tau|m, \theta)$ w.r.t. $\theta$:

$$\frac{\partial}{\partial \theta} \log p(\tau|m, \theta) = \frac{\partial}{\partial \theta} \left[ \log \left( \eta(s_1) \prod_{t=1}^{T} P(s_{t+1}|s_t, a_t) \right) + \sum_{t=1}^{T} f_\theta(s_t, a_t, m) - \log Z(\theta) \right] \tag{16}$$

$$= \sum_{t=1}^{T} \frac{\partial}{\partial \theta} f_\theta(s_t, a_t, m) - \frac{\partial}{\partial \theta} \log Z(\theta) \tag{17}$$

$$= \sum_{t=1}^{T} \frac{\partial}{\partial \theta} f_\theta(s_t, a_t, m) - \mathbb{E}_{\tau\sim p(\tau|m, \theta)} \left[ \sum_{t=1}^{T} \frac{\partial}{\partial \theta} f_\theta(s_t, a_t, m) \right] \tag{18}$$

Substituting Equation (18) into Equation (15), we get:

$$\mathbb{E}_{m\sim p(m), \tau\sim p_\theta(\tau|m)} \left[ \log q_\psi(m|\tau) \left[ \sum_{t=1}^{T} \frac{\partial}{\partial \theta} f_\theta(s_t, a_t, m) - \mathbb{E}_{\tau'\sim p_\theta(\tau|m)} \sum_{t=1}^{T} \frac{\partial}{\partial \theta} f_\theta(s_t', a_t', m) \right] \right]$$

With Lemma 1, we know that when $\omega$ is trained to optimality, we can sample from $p_{\pi_\omega^*}(\tau|m)$ to construct an unbiased gradient estimation.

## C Meta-Testing Procedure of PEMIRL

We summarize the meta-test stage of PEMIRL for adapting reward functions to new tasks in Algorithm 2.

## D Graphical Model of PEMIRL

Here we show the graphical model of the PEMIRL framework in Figure 4.

---

**Algorithm 2** PEMIRL Meta-Test for Reward Adaptation

---

**Input:** A test context variable $m \sim p(m)$, a test expert demonstration $\tau_E \sim p_{\pi_E}(\tau|m)$, and ground-truth reward $r(s, a, m)$.
Infer the latent context variable from the test demonstration: $\hat{m} \sim q_\psi(m|\tau_E)$.
Train a policy using TRPO w.r.t. adapted reward function $f_\theta(s, a, \hat{m})$.
Evaluate the learned policy with $r(s, a, m)$.

---

Figure 4: Graphical model underlying PEMIRL.

## E Ablation Studies

In this section, we perform ablation studies on the sensitivity of the latent dimensions, importance of the mutual information loss ($\mathcal{L}_{\text{info}}$) term, and stochasticity of the environment. We conduct each ablation study on the Point-Maze-Shift environment to evaluation the reward adaptation performance.

**Sensitivity of the latent dimension.** We first investigate the sensitivity of different latent dimensions by running PEMIRL with latent dimension picked from $\{1, 3, 5\}$ on Point-Maze-Shift where the ground-truth latent dimension is 3. The results are summarized in Table 3. We can observe that PEMIRL with various latent dimension specifications all outperform the best baseline (return -28.61) stably and is hence robust to dimension mis-specifications.

**Importance of $\mathcal{L}_{\text{info}}$.** As shown in Table 4, the reward function learned by PEMIRL without the mutual information objective failed to induce a good policy in the reward adaptation setting, which demonstrates the importance of using $\mathcal{L}_{\text{info}}$.

**Performance on stochastic environment.** We create a stochastic version of Point-Maze-Shift (maze size: $60 \times 100$ cm) by changing its deterministic transition dynamics into a stochastic one. Specifically, $p(s_{t+1}|s_t, a_t)$ is now realized as a Gaussian with standard deviation being 1 cm. As shown in Table 5, the average return of PEMIRL outperforms the best baseline Meta-IL by a large margin.

## F Additional Experimental Details

### F.1 Network Architectures

For all methods except AIRL, $q_\psi(m|\tau)$ and $\pi_\omega(a|s, m)$ are represented as 2-layer fully-connected neural networks with 128 and 64 hidden units respectively and ReLU as the activation function.

Following [11], to alleviate the reward ambiguity problem, we represent the reward function with two components (a context-dependent disentangled reward estimator $r_\theta(s, m)$ and a context-dependent potential function $h_\phi(s, m)$):

$$f_{\theta,\phi}(s_t, a_t, s_{t+1}, m) = r_\theta(s_t, m) + \gamma h_\phi(s_{t+1}, m) - h_\phi(s_t, m)$$

Here $r_\theta(s, m)$ and $h_\phi(s, m)$ are realized as a 2-layer fully-connected neural networks with 32 hidden units.

### F.2 Environment Details

**Point-Maze**. The ground-truth reward corresponds to negative distance toward the goal position as well as controlling the pointmass from moving too fast. We use 100 meta-training tasks and 30 meta-training tasks.

| latent dim. | return |
|---|---|
| 1 | $-10.58 \pm 1.27$ |
| 3 | $-14.13 \pm 1.21$ |
| 5 | $-15.41 \pm 1.40$ |

Table 3: PEMIRL is robust to latent dimensions.

| method | return |
|---|---|
| PEMIRL w/o MI | $-39.24 \pm 3.48$ |
| PEMIRL | $-14.13 \pm 1.21$ |

Table 4: The MI term is important for training PEMIRL.

| method | return |
|---|---|
| Meta-IL | $-30.58 \pm 4.17$ |
| PEMIRL | $-17.39 \pm 0.84$ |

Table 5: PEMIRL excels in stochastic env.

**Ant**. The ground-truth reward corresponds to moving as far as possible forward or backward without being flipped. We have 2 tasks in this domain.

**Sweeper**. The ground-truth reward is the negative distance from the sweeper to the object plus the negative distance from the object to the goal position. We train all methods on 100 meta-training tasks and test them on 30 meta-test tasks.

**Sawyer Pushing**. The ground-truth reward in this domain is similar to Sweeper, and we also use 100 meta-training tasks and 30 meta-test tasks.

### F.3  Training Details

*Training the policy.* During training TRPO, we use an entropy regularizer $1.0$ for Point-Maze, and $0.1$ for the other three domains. We find that adding an imitation objective in PEMIRL that maximizes the log-likelihood of the sampled expert trajectory conditioned on the latent context variable inferred by $q_\psi$ with scaling factor $0.01$ accelerates policy training.

*Training the inference network and the reward model.* We train $q_\psi(m|\tau)$, $r_\theta(s,m)$ and $h_\phi(s,m)$ using the Adam optimizer with default hyperparameters.

*Scaling up the mutual information regularization.* Note that in Equation 10, $\beta$ does not necessarily need to be equal to 1. Adjusting $\beta$ is equivalent to scaling $\mathcal{L}_{\text{info}}(\theta, \psi)$. We scale $\mathcal{L}_{\text{info}}(\theta, \psi)$ by $0.1$ for all of our experiments.

*Policy and inference network initialization.* We initialize and $q_\psi(m|\tau)$ using Meta-IL discussed in Section 5 while randomly initializing the policy $\pi_\omega(a|s,m)$.

*Stabilizing adversarial training.* As in [11], we mix policy samples generated from previous 20 training iterations and use them as negatives when training the disriminator. We find that such a strategy prevents the discriminator from overfitting to samples from the current iteration.

### F.4  Data Efficiency

During meta-training, for the Point-Maze environment, it takes about 32M simulation steps to converge (similar to other methods such as Meta-InfoGAIL that takes 28M), which amounts to about 2 hours on one Nvidia Titan-Xp GPU; for the Ant environment, it takes about 13.8M simulation steps (Meta-InfoGAIL takes 12M) and about 40 hours on the same hardware (the state-action dimension is much larger than that of Point-Maze).

At meta-testing phase, the data efficiency of PEMIRL is comparable to RL training with the oracle ground-truth reward as shown in Table 6.

|  | Point-Maze-Shift | Disabled-Ant |
|---|---|---|
| RL w/ oracle reward | 4M env steps | 15M env steps |
| PEMIRL | 5.4M env steps | 18M env steps |

Table 6: Comparison on data efficiency between RL trained with reward learned by PERMIL and RL trained with oracle reward. The methods have been shown to have similar data efficiency on Point-Maze-Shift and Disabled-Ant.