[Reviews · NeurIPS 2019]

Reviewer 1



The paper identifies the unsolved problem of meta-Inverse Reinforcement Learning. That is, learning a reward function for an unseen task from a single expert trajectory for that task, using a batch of expert trajectories for different but related tasks as training data (the task being solved by each training expert trajectory is not communicated to the learning algorithm). Because IRL is used rather than imitation learning, a reward function is learned for each task (or rather a single reward function parameterized by the latent variable m which is supposed to capture task). The paper then formulates an framework for training neural networks to solve the identified problem, building off of past work on Adversarial IRL, and adding latent task variables to handle the variation in task. A network q_psi is used to identify the task variable from a demonstration. The paper identifies an objective function for solving the problem, which includes a term that encourages high mutual information between the latent task variable and the trajectories induced by the reward function. They also derive a tractable sample-based method for estimating gradients of this objective. Overall I enjoyed this paper. It feels polished and complete, and the writing is quite good aside from a few awkward sentences. The problem tackled and the proposed solution are moderately novel, being a reasonably straightforward combination of past work on meta-learning and inverse reinforcement learning. The significance is high given the popularity of meta learning and IRL. One wonder I have is the importance of the mutual information term. On line 114 the paper states: """ Without further constraints over m, directly applying AIRL to learn the reward function could simply ignore the context variable m. So we need to explicitly maximize the “correlation” between the reward function and the latent variable m. """ Has this been empirically verified? It would seem to be in the interest of the discriminator to make use of m since it makes life harder for the generator, and if the discriminator is using m then the generator would have to as well. Therefore it would seem possible to omit the mutual information term, which would simplify the framework somewhat. Gradients would then, of course, have to be backpropagated through q_\psi in training both the generator and the discriminator. This starts to look like a variational autoencoder for trajectories, with the GAN taking the place of the VAE decoder. Might be interesting to test against this kind of architecture to verify the contribution of the MI. The experimental methodology seems sound for the most part. In some ways the baselines seem a bit weak; for example, none of them are really properly equipped to solve the manipulation performed in Section 5.2. But assuming this really is the first method to tackle non-tabular meta-IRL, there is little other choice. I have a few misgivings with respect to the coverage of the experiments. One is the focus on the deterministic environments. I would be interested to see performance on a stochastic environment, since this would make expert trajectories less informative about the goal that the expert is attempting to reach, and might make one-shot imitation/irl much more difficult; also, Lemma 1 and Theorem 1 both assume deterministic environments. In that case I would also be interested to see whether, for a single test task, performance improves as more expert trajectories are observed for that task. Another worry I have is that the generalization abilities required by the environments tested are fairly minimal. E.g. in point mass, the tasks seen at test time are presumably quite similar to some of the tasks seen at training time, and this applies to other envs as well. In the Ant env it seems no generalization ability is required at all since there are only two tasks. It would be interesting to evaluate this method in more realistic environments where generalization can be tested more thoroughly. Finally, I'm not completely sure about this but I think the dimensionality of the latent variables was set equal to the true dimensionality of the task space (i.e. 2 dimensions for all envs tested). What happens to performance if this dimension is misspecified? Typos: Eq 6, should be τsimpθ(τm), rather than τsimp(τm,θ). (See also eq 13 in supplemental and several other places) Eq 9, (\tau) should not be in subscript. Line 224: imitation Line 274: imitation Supp line 214: one of these should be "meta-testing" Supp line 426: missing a word, or added an "and"

Reviewer 2



Post-rebuttal: Thank you for the responses they were very useful in assessing the work better. ------ The authors present a new method to learn disentangled reward functions in a meta IRL setting. They heavily build on prior work. Mainly they utilize AIRL [11] and ideas from [23] and [32] in order to do so. [11] presents a method to infer reward functions in a IRL setting but focuses on single tasks. [23] and [32] use context variables so that policies are conditioned on tasks. In this paper the authors have combined these approaches so that disentangled rewards can be inferred in a meta IRL setting. In general I liked reading the paper. It is well written and structured and the contributions are clear. Regarding contributions, it seems to be a bit incremental since the paper doesn’t really build upon the ideas but mostly combines them in a nice way. That doesn’t take away from what the final solution offers although I do have some questions regarding the results which I will talk about later. I liked the formulation on MaxEnt IRL and MI regularization by putting constraints on m. It is a nice idea which makes sense. Since a few methods are combined here I would have liked to see some sort of discussion regarding the sensitivity of the approach. The same goes for the constraints put on MI, \alpha and \beta, how do these values change the learning process and how hard is it to tune. On a similar note there's no talk on data efficiency. How long does it take for agents to train on the new task. I can understand that recovering reward functions allows agents to transfer and learn through trial and error, but there needs to be a discussion on what that constitutes. In the reward adaptation, the agent in PEMIRL continues learning. For how long, is that better than straight up learning from scratch in the new task with new environment dynamics? In the supplementary material in the testing algorithm it is mentioned ”Train a policy using TRPO w.r.t. adapted reward function” but no further information is provided. Such a comparison would provide some more insight, rather than showing only results for which other methods completely fail, which was also expected for some of them since they either don’t condition on the task, or they don’t recover rewards functions and hence cannot continue learning and adapt. Regarding the results and experimental design. As I understand it, during training the only thing that changes is the reward/goal (e.g. Point-Maze Navigation). I wonder what the case would be if during training, the tasks could also change the dynamics e.g. have the wall be at random places. I would think that in that case methods what utilize context variables should be able to learn context conditioned policies, even if they don't recover the reward since conditioning would also take into account environment dynamics to a certain extent. That could potentially provide better generalizations. To elaborate, in Meta-IL meta learning happens conditioning on the context variable. However the tasks given only change wrt reward while they dynamics stay fixed. It is natural that this will not benefit this algorithm and it will fail. I think that if during training the environment dynamics were also changing then it would make sense that the context variable learned would be able to better find policies matching the task. I don’t think the results speak the full truth about how much better PEMIRL is compared to approaches that do not recover the reward function for this reason. I feel that what is provided during training plays an important part which is not fully explored here. Which brings me to unstructured demonstrations. I was confused about the unstructured aspect. I assume what this means is that the task distribution does not have to conform to constraints put by the learning algorithm? E.g. If the algorithm needs changing dynamics in its training data? If that is the case, then it is clear why Meta-IL doesn’t work. But it seems to me providing tasks that only change the goal/rewards (as I understand is happening during training) is also a bit structured. Perhaps a fairer comparison would have been to change everything? Or perhaps provide a mix or both changing goal/fixed dynamics, static goal/changing dynamics. Minor: “imitaiton” in multiple places. Table 2. I don't understand what the policy generalization refers to. I understand the reward adaptation part, but the policy generalization confuses me? Is there something I am missing? Is this for the disabled ant experiment. I think the caption should be a lot more informative than that. 198: “our method is not restricted to discrete tabular settings and does not require access to grouped demonstrations sampled from a task distribution”. I found this a bit weird. The paper builds on methods in [11][23][32] which are in continuous tasks of similar complexity as in this research. 284: “PEMIRL is the first model-free Meta-IRL algorithm that can achieve this and scale to complex domains with continuous state-action spaces”. I find this a bit strong. [23][32] also have been tested in complex continuous domains. Personally I would remove this.

Reviewer 3



The submission studies the meta-inverse reinforcement learning problem, which is clearly motivated. Specifically, the submission includes a latent variable into the AIRL. To solve the achieved deep latent variable model, a new estimation method is proposed to approximately optimize the objective function. The work is very complete with a new model, a corresponding estimation method, theoretical supports, and convincing experiments. One minor issue: Authors mentioned that the model is related to a unified graphical model (line 55) in the introduction, which,however, is never mentioned in the main part.

[Author Response · NeurIPS 2019]

| latent dim. | return |
|---|---|
| 1 | $-10.58 \pm 1.27$ |
| 3 | $-14.13 \pm 1.21$ |
| 5 | $-15.41 \pm 1.40$ |

| method | return |
|---|---|
| PEMIRL w/o MI | $-39.24 \pm 3.48$ |
| PEMIRL | $-14.13 \pm 1.21$ |

Table 1: PEMIRL is robust to various latent dimensions.

Table 2: The MI term is important for training PEMIRL.

Figure 1: Graphical model underlying PEMIRL

We thank all the reviewers for the constructive feedback. We will incorporate the valuable suggestions in the revised
version. We have conducted more experiments and addressed all of the comments below:

**To Reviewer #2:**

**Q1: The importance of mutual information (MI) term?** We conducted an ablation study on the MI term with the
Point-Maze-Shift environment. The reward function learned without MI failed to induce a good policy in the reward
adaptation setting. Results in Table 2 (on the top) demonstrates the importance of our MI term.
Theoretically, without MI regularization, the resulting method indeed resembles a VAE. As analyzed in [36], the ELBO
of VAE can be interpreted as enforcing consistency between $p(m)p_\theta(\tau|m)$ and $p_E(\tau)q_\psi(m|\tau)$ by minimizing the KL
divergence between these joint distributions. Without maximizing the MI between $m$ and $\tau$, a simple degenerate case is
$p_\theta(\tau|m) = p_E(\tau)$ and $q_\psi(m|\tau) = p(m)$, which satisfies the consistency constraints, yet completely fails to capture the
dependencies between $m$ and $\tau$.

**Q2: What if latent dimension is mis-specified?** We conducted additional experiments with the Point-Maze-Shift
environment (where the ground-truth latent dimension is 3). See the results in Table 1 (on the top). We can observe
that PEMIRL with various latent dimension specifications all outperform the best baseline (return -28.61) stably and is
hence robust to dimension mis-specifications.

**Q3: Performance on a stochastic environment?** We create a stochastic version of Point-Maze-Shift (maze size:
$60 \times 100$ cm) by changing its deterministic transition dynamics into a stochastic one. Specifically, $p(s_{t+1}|s_t, a_t)$ is
now realized as a Gaussian with standard deviation being 1 cm. The average return of PEMIRL in reward adaptation is
$-17.39 \pm 0.84$, which outperforms the best baseline (average return $-30.58$) by a large margin.

**Q4: Test generalization in more realistic environments?** We will add an experiment with a simulated Sawyer robot
button pressing task to the revised version, which we were unable to complete during the rebuttal period.

**To Reviewer #3:**

**Q1: Discussion on data efficiency?** We would like to clarify that in reward adaptation, we use the inferred reward
function to train a policy from scratch rather than finetuning the learned policy. Although efficiency is not the focus of
this work, we are happy to provide more discussions on this aspect in the revised version. The sample complexity of
PEMIRL at meta-testing phase is comparable to RL training with the oracle ground-truth reward, *e.g.* (PEMIRL vs RL
with oracle reward): Point-Maze-Shift: 5.4M vs 4M simulation steps; Disabled-Ant: 15M vs 18M simulation steps.

**Q2: Can the tasks also change the dynamics during training?** In principle, our algorithm can also handle changes
in dynamics during meta-training. We leave this as an interesting avenue for future work.

**Q3: The meaning of unstructured demonstrations?** As described in line 58-59, "unstructured" means the demon-
strations are not grouped according to the task or labeled by task-specific variables. To elaborate, as discussed in line
196-199, previous Meta-IRL methods [12, 32] make simplifying assumptions that each provided expert demonstration
contains its corresponding task information (hence "structured"), while PEMIRL has to learn to infer the underlying
task corresponding to each demonstration. We will rephrase corresponding parts to clarify it in the revised version.

**Q4: Minor comments (1)** We will revise the captions to make them more informative. Policy generalization examines
if the policy learned by Meta-IL is able to generalize to new tasks with new dynamics. **(2 & 3)** [11, 23] focus on
standard IRL and meta-RL respectively rather than Meta-IRL as in PEMIRL. Although [32] focuses on Meta-IRL, their
method derivation (*e.g.* Eq 5) requires a tabular MDP. We will rephrase corresponding parts to make this clear.

**To Reviewer #5**

**Q1: Discussion on the efficiency of the proposed method?** Although efficiency is not the focus of this work, we are
happy to provide more discussions on this aspect in the revised version. During meta-training, for the Point-Maze
environment, it takes about 32M simulation steps to converge (similar to other methods such as Meta-InfoGAIL
that takes 28M), which amounts to about 2 hours on one Nvidia Titan-Xp GPU; for the Ant environment, it takes
about 13.8M simulation steps (Meta-InfoGAIL takes 12M) and about 40 hours on the same hardware (the state-action
dimension is much larger than that of Point-Maze). For the sample complexity of meta-testing phase, please refer to the
response to Q1 for reviewer #3.

**Q2: Graphical model illustration?** We will add the graphical model illustration in Figure 1 to the revised version.

[Meta-Review · NeurIPS 2019]

The reviewers agree that the paper is interesting and a good contribution. Some suggestions for the final version: - spell checker - Have a look at the meta-RL literature that also deals with latent variables: http://auai.org/uai2018/proceedings/papers/235.pdf https://openreview.net/forum?id=rk07ZXZRb